# LegoNet: Piecing Together and Breaking Apart Sub-Networks for Scalable Multi-task Learning

## Abstract

Despite considerable progress in general-purpose vision models, most efforts focus on designing a new unified structure that can handle different types of input and supervision. In contrast, we believe each vision task requires its specific designed module to use different forms of perception. For example, a feature pyramid network is commonly used in segmentation but not in classification. We present LegoNet, a general Multi-Task Learning (MTL) framework that is assembled with many small sub-networks from different vision tasks, similar to how Lego pieces can be pieced together into larger structures. By leveraging this property, LegoNet can borrow design elements from single-task models and combine them to create a scalable multi-task model. We demonstrate its efficiency on mainstream vision datasets such as ImageNet, COCO, and ADE20K, and show it achieves comparable results to state-of-the-art single-task models. Moreover, like a Lego creation capable of dynamically piecing together or breaking apart pieces, our model exhibits scalability in both its model capacity and adaptability to a multitude of tasks. It can remove sub-networks and decompose into high-performing components for efficient adaptation, or add sub-networks for learning new tasks in a continuous learning scenario. On downstream tasks, it can be fine-tuned with fewer training parameters, fewer model parameters, and even transformed to a low computation shape. These functions can be controlled and combined to meet various demands of downstream applications.

## 1 Introduction

Comprehensive visual understanding demands a general-purpose model capable of performing diverse vision tasks. With a similar goal, multitask learning (MTL), which enables the simultaneous training of models on multiple tasks and allows them to leverage shared information, has been explored extensively. Recently, some efforts (Jaegle et al., 2021; Lu et al., 2023; Wang et al., 2022) have been made on unified input and supervision of the vision tasks so a single large model can be trained to perform multiple tasks. As such a large model has the advantage of training on a large and diverse set of data, it often requires new designs to incorporate all the vision tasks and often fails to benefit from the existing SoTA single-task model design. For example, Unified-IO (Lu et al., 2023) treats detection as a language modeling task to regress the bounding box location and discard the traditional detector design. This particular design enables to building of a general framework for various tasks while sacrificing the performance of individual tasks.

To address this challenge, we developed LegoNet, a versatile framework that enables the seamless integration of various sub-networks from different vision tasks without requiring any new design elements such as a unified representation. Our objective was to develop a general multi-task framework that could be easily assembled from several single-task models and directly incorporate a variety of tasks. LegoNet achieves this by utilizing a mixture-of-experts (MoE) vision transformer as its backbone and incorporating task-specific sub-networks from each task. Each MoE module has several experts (also referred to as sub-networks) that have the same shape as in a single-task backbone model. The weight of each sub-network including backbone and task heads can be trained from scratch or directly loaded from a pre-trained single-task model. When using pre-trained weights, it takes only 1/2 of the training epochs compared to training from scratch. This approach enables us to

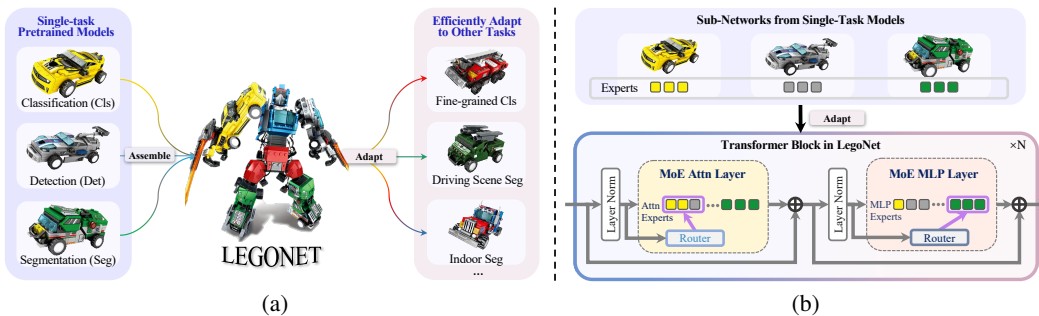

Figure 1: **Overview of LegoNet.** (a) LegoNet is a general multi-task learning (MTL) framework that is assembled with many small sub-networks from different vision tasks similar to how Lego pieces can be assembled into larger structures. These sub-networks can be further selected and utilized on downstream tasks. (b) It shows how LegoNet load weights of sub-networks from single-task models.

efficiently combine sub-networks from different tasks while maintaining a smooth learning process with high performance.

As shown in Figure 1, we constructed our multi-task framework by assembling sub-networks from individual tasks. Our results demonstrate that stacking sub-networks from single-task models is a robust approach for multi-task learning and can achieve results comparable to state-of-the-art single-task models on mainstream vision datasets. By showcasing its scalability to handle additional tasks, we provide evidence of LegoNet's ability to adapt and effectively tackle diverse vision tasks. This is non-trivial as it highlights the framework's potential to serve as a flexible and powerful solution for addressing a broad spectrum of vision-related challenges.

There are two main advantages of our framework compared to other large-scale multi-task models. The first advantage is its ability to easily attach or detach sub-networks, much like adding or removing pieces in a Lego construction. This is particularly significant when incorporating more tasks into our multi-task framework dynamically, such as in a continuous learning scenario. By attaching sub-networks, we can efficiently adapt to new tasks and datasets. Conversely, detaching sub-networks can be used to prune the model and meet memory limitations when deploying on downstream tasks.

Another advantage of the LegoNet is its ability to quickly and flexibly adapt to downstream tasks. This is made possible by the mixture-of-experts module, which allows the model to select the most semantically meaningful sub-networks for faster transfer to downstream tasks (See Sec3.4). LegoNet can automatically select some of the sub-networks to fine-tune and freeze the rest of the framework, reducing computation costs by activating fewer sub-networks in a forward pass. While large-scale models benefit from their capacity, downstream tasks may not require all of this capacity, and automatic selection can be beneficial in reducing unnecessary computation.

Overall, the LegoNet framework offers a versatile and efficient solution for multi-task learning, with the ability to easily adapt to new tasks and efficiently utilize resources. Its fast and flexible adaptation to downstream tasks makes it a valuable tool for the vision community. Our main contributions can be summarized as follows:

- **A versatile multi-task model, assembled from individual single-task models, excels at robustly addressing various vision tasks.** LegoNet achieves performance on par with state-of-the-art single-task models across all tasks. Extending our framework to additional tasks is effortless: simply append sub-networks and expedite training by leveraging pre-trained weights from a single-task model.

- **Adaptive Model Size Adjustment.** Our framework exhibits a dynamic scaling property, allowing the addition or removal of sub-networks at any training stage. This flexibility and adaptability hold significant value, particularly within the context of scalable Multi-Task Learning.

- **Efficient adaptation on downstream tasks.** LegoNet enables versatile architectural control, offering several direct and efficient methods for tailoring the structure.

- **Continual learning without forgetting.** The model can effortlessly leverage existing sub-networks to adapt to new tasks by learning new routers. Additionally, it integrates new sub-networks without disrupting the current architecture, preventing catastrophic forgetting.

Figure 2: **Efficient adaptation with dataset-specific routing.** Left: Dataset-specific routers select distinct experts for individual datasets. Middle up: Reduce model parameters by learning new routers and eliminating infrequently selected experts. Middle down: Simple model expansion via adding and training a handful of fresh experts per MoE module while keeping existing experts freezing. Right up: Reduce training parameters by exclusively learning new routers and a few optional experts while preserving other parameters. Right down: Minimize computational costs by training new routers with a smaller Top-K, resulting in fewer experts being selected in a single forward pass. These adaptation techniques can be combined to meet specific requirements.

## 2  RELATED WORK

**Multi-task Learning.** Multi-task learning (Kendall et al., 2018) jointly learns multiple related tasks with a single model. Recently, transformer-based MTL architectures (Xu et al., 2022) have gained popularity. Some works (Jaegle et al., 2021; Lu et al., 2023; Cai et al., 2022) attempt to unify the input and output space for different tasks. Others (Chen et al., 2023; Xu et al., 2022; Maninis et al., 2019; Kokkinos, 2017) remove complicated task-specific modules for simplicity and conduct multi-task learning on a multi-label dataset. In contrast, LegoNet is a versatile multi-task framework that can seamlessly integrate additional tasks by assembling sub-networks derived from single-task models. While it maintains architectural similarities to common multi-task models with a shared backbone and distinct task heads, its key distinction lies in the remarkable flexibility to affix or remove sub-networks and its proficiency in efficient adaptation.

**Mixture of Experts (MoE).** Jacobs et al. (1991) introduced the MoE to merge sub-networks and perform conditional computation. Recently, this technique has been used to reduce computation while maintaining model capacity (Shazeer et al., 2017). Some studies (Lepikhin et al., 2021; Fedus et al., 2022; Riquelme et al., 2021; Mustafa et al., 2022) have leveraged MoE to train models with trillions of parameters with relatively low computation. Mod-Squad (Chen et al., 2023) and M3ViT (Xu et al., 2022) also use MoE in their MTL model to enhance optimization and performance. In contrast, our main use of MoE is for sub-network management, including adding, removing, and selecting experts for downstream tasks.

**Parameter-efficient transfer learning.** The Adapter technique was proposed as a standalone layer that can be integrated into an existing neural network for efficient transfer. LoRA (Hu et al., 2021) utilizes a bottleneck structure to enforce a low-rank constraint on the weight updates. Other approaches integrate CLIP-based adapters (Gao et al., 2021; Yi-Lin Sung, 2022; Zhang et al., 2021), upsampling and downsampling modules (Li et al., 2022), and additional bias parameters (Zaken et al., 2022) to reduce training parameters during fine-tuning. Our research, on the other hand, centers on the precise selection of the most semantically relevant model components and efficient adaptation to downstream tasks, all without necessitating the creation of additional, bespoke modules.

**Dynamic network.** Dynamic neural networks, which can adapt their structures during inference, have shown notable improvements in computational efficiency compared to static models (Han et al., 2022). Previous works have focused on adjusting the network depth (Bolukbasi et al., 2017; Veit & Belongie, 2018; Wang et al., 2018; Huang et al., 2018) or width (Yuan et al., 2020; Li et al., 2021; Yu et al., 2019), or conducting dynamic routing within a fixed supernet including multiple

possible paths (Li et al., 2020; Liu & Deng, 2018). However, dynamic networks require training for architectural adjustments, mainly targeting pre-trained datasets. Post-training architecture changes and downstream task adaptation are challenging. In contrast, LegoNet enables straightforward expansion to new tasks and streamlined adaptation to downstream tasks, offering a versatile approach in comparison to traditional dynamic networks.

**Continual learning.** Continual learning involves handling a diverse set of tasks and accumulating knowledge through a series of training. Recent efforts have been made to address catastrophic forgetting, including imposing regularization (Kirkpatrick et al., 2017; Zenke et al., 2017; Ritter et al., 2018) and retaining a small buffer of data for replay (Lopez-Paz & Ranzato, 2017; Nguyen et al., 2018). Some approaches (Yoon et al., 2018; Hung et al., 2019) dynamically expand the network by adding neurons to each MLP or convolution layer. In contrast, LegoNet offers a straightforward, well-structured expansion approach by attaching new sub-networks and learning new routers. Notably, as each dataset has its own router, the previously added sub-networks are unaffected by the new dataset. Unlike alternative expansion techniques, our approach avoids catastrophic forgetting.

## 3 METHOD

### 3.1 DEFINITION AND PREREQUISITE

**Problem definition.** Unlike common MTL on a single image set with multiple task labels, our framework is trained on combinations of single-task datasets, which is similar to previous work (He et al., 2022; Ghiasi et al., 2021). We say *heterogeneous* to refer to the combination of single-task datasets. We start with the definition of multi-task heterogeneous training. Suppose we have $M$ datasets $D_1, D_2, ..., D_M$. Each dataset contains a set of training pair $\{I; T_i(I)\}$ and $T_i$ is the task on dataset $D_i$ that map images $I$ to $T_i(I)$. Here, we assume each dataset only has one task to do for simplicity. Multi-task heterogeneous training is to learn a joint model on the $M$ datasets at once.

**Mixture-of-Experts (MoE).** A MoE layer contains a group of expert networks $E_1, E_2, ..., E_N$ and a routing network $G$. The routing network $G$ calculates the weight $G^k(x)$ for each expert $E_k$ given input $x$ and the output of an MoE layer is the weighted sum of the output of every expert $E_k(x)$. Formally, the output of an MoE layer is

$$y = \sum_{k=1}^{N} G^k(x) E_k(x). \tag{1}$$

The routing network $G$ is a Top-$K$ Routing network (Shazeer et al., 2017) that only $K$ experts with the highest weight contribute to the final output:

$$G(x) = \text{TopK}(\text{Softmax}(xW_g), k) \tag{2}$$

where $\text{TopK}(\cdot, k)$ zeroes all vector elements except the elements with the largest $K$ values.

**Mutual information (MI) loss.** Mod-Squad (Chen et al., 2023) proposes an MI loss as an auxiliary loss to better assign experts to *tasks* so that each expert is more likely to be used for a fixed set of tasks. In contrast, the key motivation in LegoNet is to encourage experts to specialize on *datasets* and then when adapting to downstream tasks, the downstream datasets are more likely to activate a small subset of experts. So we have $M$ *dataset-specific routing networks* and modify the loss so that the experts are assigned to datasets instead of tasks:

$$L_{MI} = -\sum_{i=1}^{M} \sum_{j=1}^{K} P(D_i, E_j) \log P(D_i, E_j) + \sum_{i=1}^{M} P(D_i) \log P(D_i) + \sum_{j=1}^{K} P(E_j) \log P(E_j). \tag{3}$$

As in (Chen et al., 2023), we assume that $P(D_i) = \frac{1}{M}$ as we want all datasets to be considered equally important. We have $P(E_j|D_i) = \sum_{x \in D_i} G_i^j(x)$ where $G_i^j$ is the weight of expert $E_j$ for dataset $D_i$. With $P(E_j|D_i)$, we can get $P(D_i, E_j) = P(E_j|D_i)P(D_i)$ and $P(E_j) = \sum_{i=1}^{M} P(D_i, E_j)$.

### 3.2 LEGONET

LegoNet is a framework orthogonal to any transformer-based single-task architecture. We use a mixture-of-experts (MoE) vision transformer as backbone (Fig. 1). We replace the attention and MLP

layers in a transformer block with MoE Attention and MoE MLP layers. On top of the backbone, we directly incorporate all task-specific designed modules (e.g., feature pyramid network).

**Load weight from pre-trained single-task models.** We assume that LegoNet and single-task models use the same network architecture (e.g., MoE Swin Transformer). The only difference is that LegoNet could have more experts in an MoE module. Inside an MoE module, each expert can load weight from an expert in a pre-trained single-task model as they are exactly the same shape. We explore two situations of loading weight from multiple single-task models: 1) **Full Loading**: LegoNet has enough experts in an MoE module to load from all single-task models. In this scenario, LegoNet has $M * E$ experts in each MoE module and the single-task model has $E$ experts. LegoNet can directly load from $M$ single-tasks model. 2) **Partial Loading**: LegoNet does not have enough experts to load from all single-task models' experts. In this scenario, both LegoNet and single-task models have $E$ experts in an MoE module and LegoNet can still load weights from a subset of single-task models. For each expert in LegoNet, we randomly load weight from $M * E$ experts. For both methods, they will directly load task-specific modules from single-task models and the rest of the model (e.g., linear projection of flattened patch) can be initialized randomly.

**Sampling strategy.** Multi-task models that update the network after forwarding for all tasks are impractical as GPU memory is heavily consumed when activating all dense vision modules (e.g., segmentation heads). To address this issue, LegoNet adopts a two-step sampling approach. First, we apply weighted sampling to select one out of the $M$ datasets. Then, we randomly sample a batch of data from the chosen dataset. The weight assigned to each dataset $D_i$ for sampling is denoted as $w_{sample_i}$, which can be pre-defined based on the total number of required iterations for convergence in single dataset training, with some empirical tuning.

**New mutual information (MI) loss for heterogeneous training.** In Mod-Squad (Chen et al., 2023), the MI loss in Eq. 3 can be calculated in each batch as all tasks are contained in one batch. However, calculating $P(D, E)$ and $P(E)$ within a sampled batch from one random dataset in heterogeneous training leads to heavy bias. To address this, we use an approximation inspired by :

$$\frac{\partial}{\partial x}[x \log x] = 1 + \log x = \frac{\partial}{\partial x}[(1 + \log c)x]|_{c=x}.$$  (4)

This suggests that if we replace $x \log x$ with $(1 + \log c)x$, and $c$ is a good approximation of $x$, then we will still have a similar gradient. In our case, we will approximate a *running estimate* of the joint distribution of $P(D, E)$ with a buffer $B(D, E)$. The running estimate $B(D, E)$ avoids the heavy bias caused by estimating $P(D, E)$ from a single task data set. In each forward pass when we sample dataset $D_i$, we momentum update $B(D_i, E)$ with a momentum of 0.98. This keeps the estimate of $B$ close to that of the desired joint distribution. Using this idea, we rewrite Eq. 3 and use the resulting equation as the loss function to calculate the gradient. The equation is given by:

$$L_{MI} = -\sum_{i=1}^{M}\sum_{j=1}^{K}[1 + \log B(D_i, E_j)]P(D_i, E_j) + \sum_{j=1}^{K}[1 + \log(\sum_{i=1}^{M} B(D_i, E_j))]P(E_j).$$  (5)

Here, $P(D_i, E_j), P(E_j)$ is calculated in each forward pass backpropping gradients. If $D_i$ is not sampled in the current forward pass, $P(D_i, E_j)$ is set to 0. Note that $P(D_i) \log P(D_i)$ is ignored as a constant. When adapting to new downstream datasets, the buffer still memorizes $P(D, E)$ for old datasets. Therefore, the MI loss can still be computed to balance experts on new datasets, which is not applicable in (Chen et al., 2023).

### 3.3 DYNAMIC SCALING PROPERTY IN MODEL SIZE

**Scale down on downstream tasks.** As shown in Fig. 2, LegoNet has dynamic scaling property in model size. When adapting to an existing/new task, LegoNet can learn a new router for the task while freezing the rest of the parameters to remove experts that are rarely selected by this task-specific router. This property is particularly useful for scaling down a large pre-trained model to better fit a new task with improved efficiency.

**Scale up for continuous learning.** LegoNet is capable of simple model expansion that can help conduct continual learning. Specifically, we directly add $C$ experts in each MoE module along with new task-specific routers every time we learn a new task. All parameters except for the newly added part are frozen during training. There are three main advantages of this approach: 1) No catastrophic

forgetting. As all the experts are unchanged after learning and the newly added experts will not be chosen by the router of previous tasks, there is no catastrophic forgetting. 2) Well-organized architecture and knowledge reuse. The model maintains an elegant architectural design. The routers select experts to reuse knowledge related to the new task and ignore experts with unrelated expertise. 3) The computation cost is constant. Other expanding methods Yoon et al. (2018); Hung et al. (2019) add both computation cost and capacity to the existing model, while our approach only adds capacity. This makes our approach expandable with a large number of tasks.

### 3.4 EFFICIENT ADAPTATION ON DOWNSTREAM TASKS

LegoNet offers two appealing advantages: **1) Downstream applications can select the best-matching experts for the task at hand**, similar to how a transformer transfers to different cars in Fig. 1. This can be done by learning a new router in each MoE module to find good experts for the downstream task. We consider an expert as a good expert if it is chosen with a high frequency by the router on the downstream dataset. The routers are very lightweight (0.4M in parameters) and can quickly converge to the optimum while freezing all other parameters. **2) We can easily control the architecture within each MoE module.** The model can be expanded or pruned by adding or removing experts, and the number of activated experts can be controlled by learning new routers with different Top-K. This flexibility enables efficient customization of the model based on the specific requirements of the task at hand.

With these two advantages, we achieve 3 types of efficient fine-tuning (see Fig. 2). **1) fewer training parameters**. The model only needs to learn a new router for the downstream dataset and optionally fine-tune a few experts in each MoE module. **2) fewer model parameters**. Same as the scale-down property we introduced before. **3) Less computation**. The new router for the downstream dataset can be learned with a smaller Top-K. So that fewer experts are chosen during one forward pass and can greatly reduce the computation cost and inference latency. Note that all these ways of efficient adaptation can be combined together to meet the demands of downstream datasets.

## 4 EXPERIMENTS

### 4.1 MULTI-TASK PRE-TRAINING

We conduct three fundamental vision tasks (classification, detection, and segmentation) on ImageNet-1K (Deng et al., 2009), COCO (Lin et al., 2014), and ADE20K (Zhou et al., 2017b). For downstream performance, we evaluate classification on the scene dataset Places-365 (Zhou et al., 2017a) (P365), the popular fine-grained dataset iNaturalist-2018 (Van Horn et al., 2018) (iNat18), the pet dataset Pets (Parkhi et al., 2012), the fine-grained bird dataset CUB (Wah et al., 2011), and the car dataset Cars (Krause et al., 2013). We evaluate downstream detection on PASCAL VOC (Everingham et al., 2010) and segmentation on Cityscapes (Cordts et al., 2016) and NYU (Silberman et al., 2012).

**Models and baselines.** We utilize Swin Transformer (Liu et al., 2021) and DaViT (Ding et al., 2022) as our backbone transformers, with different sizes: tiny (T), small (S), and base (B). Each task has its own task-specific head. For classification, we use a single linear layer. For detection, we use the retina head (Lin et al., 2017). For segmentation, we use the UperNet (Xiao et al., 2018). Each task follows its own input and output format based on single-task methods. We implement our methods and baselines as the following: 1) Train from scratch (Scratch): a vanilla single-task learning baseline that trains from scratch. 2) Pre-train then fine-tune (Pre. & FT.): pre-training on ImageNet followed by fine-tuning on the target dataset. 3) Hard sharing: Multi-task learning with multiple task-specific heads and a shared backbone (w/o MoE module) between all tasks, which is a common method in the literature. 4) LegoNet.

**Configs.** We employ 12 experts with Top-K as 4 for all MoE modules, following Chen et al. (2023). All models are trained for 240k iterations on 96 Tesla V100 GPUs with Lamb optimizer (You et al., 2019). Data augmentations for each task follow the common practice in SwinT and DaViT. For a fair comparison, all results are obtained from our implementations with the same settings. More details of the training can be found in the supplementary.

**Multi-task pre-training.** We compare different training schemes as shown in Tab. 1. Across all three datasets with varying backbones, we observe that: 1) LegoNet performs on par with the state-of-the-art

Table 1: **Multi-task pre-training.** We compare it with training from scratch (scratch), pre-training then fine-tuning (pre. & ft.), and Hard-Sharing. On COCO and ADE20K, pre. & ft. would initialize the backbone with an IN-1K pre-trained model. The numbers of parameters and FLOPs of the backbone are measured. Classification has input resolution as $224 \times 224$.

| Backbone | Model | Params (M) | FLOPs (G) | Iters (K) | IN-1K top-1 | COCO mAP | $mAP_{50}$ | $mAP_{75}$ | ADE20K mIoU | $mAcc$ | $aAcc$ |
|---|---|---|---|---|---|---|---|---|---|---|---|
| Swin-T | Scratch | 27.5×3 | 4.4 | 165 | **80.6** | 34.9 | 54.3 | 36.6 | 32.0 | 41.4 | 75.8 |
| | Pre. & FT. | 27.5×3 | 4.4 | – | – | 42.0 | 64.7 | 45.9 | 44.3 | 55.8 | 81.0 |
| | LegoNet | 50.9 | 5.1 | 240 | 80.3 | **45.0** | 66.5 | 48.2 | **44.6** | 55.0 | 81.0 |
| Swin-S | Scratch | 48.9×3 | 8.5 | 165 | **82.6** | 36.3 | 55.6 | 38.4 | 34.5 | 43.9 | 77.1 |
| | Pre. & FT. | 48.9×3 | 8.5 | – | – | **46.0** | 68.0 | 49.9 | **47.0** | 56.9 | 81.7 |
| | LegoNet | 89.1 | 9.2 | 240 | 82.0 | 45.7 | 66.8 | 49.1 | 46.7 | 57.1 | 81.8 |
| Swin-B | Scratch | 86.7×3 | 15.1 | 165 | **83.1** | 35.5 | 54.7 | 37.4 | 35.4 | 44.8 | 77.6 |
| | Pre. & FT. | 86.7×3 | 15.1 | – | – | 47.3 | 69.0 | 51.2 | 47.7 | 58.7 | 82.3 |
| | LegoNet | 158.3 | 16.2 | 240 | 82.3 | **47.6** | 69.1 | 50.9 | **48.2** | 59.0 | 82.5 |
| DaViT-T | Scratch | 27.6×3 | 4.4 | 165 | **82.5** | 37.7 | 57.1 | 40.0 | 36.4 | 46.4 | 77.8 |
| | Pre. & FT. | 27.6×3 | 4.4 | – | – | **45.4** | 66.9 | 48.4 | 45.8 | 56.0 | 81.8 |
| | LegoNet | 51.2 | 5.1 | 240 | 82.0 | 45.1 | 67.5 | 48.1 | **47.4** | 57.1 | 82.1 |
| DaViT-S | Scratch | 49.0 ×3 | 8.6 | 165 | **83.8** | 37.8 | 56.7 | 40.5 | 38.2 | 48.4 | 78.8 |
| | Pre. & FT. | 49.0 ×3 | 8.6 | – | – | **47.2** | 68.9 | 50.7 | 48.3 | 60.2 | 82.3 |
| | LegoNet | 88.9 | 9.2 | 240 | 83.3 | 46.4 | 67.7 | 49.5 | **48.6** | 59.8 | 83.0 |
| DaViT-B | Scratch | 86.9×3 | 15.2 | 165 | **84.2** | 38.0 | 57.2 | 40.5 | 38.5 | 48.7 | 78.9 |
| | Pre. & FT. | 86.9×3 | 15.2 | – | – | **48.1** | 69.7 | 51.3 | 49.3 | 60.2 | 83.0 |
| | LegoNet | 158.7 | 16.3 | 240 | 83.6 | 47.8 | 69.5 | 51.5 | **49.6** | 60.1 | 83.1 |

Table 2: **Compare of different ways of loading weight.** We report the number of iterations to show partial and full loading (Partial L. and Full L.) can speed up convergence.

| Backbone | Model | Params (M) | FLOPs (G) | Iters (K) | IN-1K top-1 | COCO mAP | $mAP_{50}$ | $mAP_{75}$ | ADE20K mIoU | $mAcc$ | $aAcc$ |
|---|---|---|---|---|---|---|---|---|---|---|---|
| Swin-T | Hard-Sharing | 27.5 | 4.4 | 260 | 79.7 | 43.8 | 65.7 | 46.8 | 44.4 | 54.8 | 80.5 |
| | Avg. L. | 27.5 | 4.4 | 260 | 79.8 | 43.6 | 65.5 | 47..0 | 44.2 | 53.9 | 80.1 |
| | Partial L. | 50.9 | 5.1 | 190 | 80.0 | 44.6 | 66.1 | 47.8 | 44.4 | 54.8 | 81.1 |
| | Full L. | 50.9 | 5.1 | **120** | 80.2 | **45.0** | 66.7 | 48.1 | **44.8** | 55.4 | 80.7 |
| | LegoNet | 50.9 | 5.1 | 240 | **80.3** | **45.0** | 66.5 | 48.2 | 44.6 | 55.0 | 81.0 |

pre-train then fine-tune learning scheme. 2) Notably, for the segmentation task, LegoNet consistently outperforms the previous state-of-the-art across all backbone choices, suggesting that joint training with classification and detection tasks improves segmentation. 3) LegoNet also works pretty well on image detection and is superior to previous arts in most cases. 4) The LegoNet and Hard-Sharing generally exhibit similar performance on tiny and base models and LegoNet consistently outperforms Hard-Sharing on tiny models, likely influenced by the relationship between model capacity and dataset scale. Finally, we want to emphasize that our framework prioritizes flexibility and adaptability over multi-task learning performance.

**Load weight from pre-trained single-task models.** We explore Full and Partial Loading in Tab. 2. Our experiments were conducted using 12 experts in each MoE module. Full loading involved loading weights from single-task models with 4 experts, while partial loading involved loading weights from single-task models with 12 experts. Full loading can save 50% of training iterations, while partial loading can save approximately 15% without compromising performance. Additionally, we compared our results with average loading (Avg. L.) based on hard-sharing, where the weights from three single-task models are averaged. This method did not speed convergence.

**Downstream performance.** As shown in Tab. 3, we compare with several methods on the downstream datasets. LegoNet outperforms the single-task pre-trained model IN-1K Pre and the multi-task model Mod-Squad, with a significant improvement in detection and segmentation. Additionally, LegoNet consistently outperforms Hard-Sharing, which suggests that adding more experts for selection could be advantageous for downstream tasks.

Table 3: **Comparisons of different MTL methods on downstream performance.** We compare with IN-1K pre-trained model (IN-1K Pre.), multi-task model Mod-Squad trained on Taskonomy (Zamir et al., 2018), and Hard-Sharing learned on our training datasets. To calculate the mean, we first average the performance on classification, detection, and segmentation separately. Afterward, we average the results across all tasks.

| Backbone | Method | P365 top-1 | iNat18 top-1 | Pets top-1 | CUB top-1 | Cars top-1 | PASC. $mAP$ | City. $mIoU$ | NYU $mIoU$ | **Mean** |
|---|---|---|---|---|---|---|---|---|---|---|
| Swin-B | IN-1K Pre. | 58.7 | 72.9 | 94.0 | 83.9 | 94.0 | 76.9 | 80.6 | 76.2 | 78.7 |
| | Mod-Squad (Chen et al., 2023) | 56.4 | 69.4 | 92.3 | 79.8 | 93.7 | 77.2 | 81.1 | 77.5 | 78.1 |
| | Hard-Sharing | 59.1 | 73.3 | 94.2 | 84.3 | 94.2 | 78.7 | 82.1 | 78.0 | 79.9 |
| | LegoNet | 59.4 | 73.6 | 94.6 | 84.7 | 94.9 | 79.1 | 82.5 | 78.7 | **80.4** |
| Davit-B | IN-1K pre. | 59.2 | 73.4 | 94.4 | 88.4 | 94.9 | 77.4 | 81.5 | 76.7 | 79.5 |
| | Hard-Sharing | 59.6 | 73.5 | 94.8 | 89.0 | 95.0 | 78.8 | 82.7 | 78.6 | 80.6 |
| | LegoNet | 60.1 | 73.9 | 94.9 | 89.4 | 95.0 | 79.5 | 83.4 | 79.3 | **81.2** |

Table 4: **Efficient adaptation.** All experiments use LegoNet as the pre-trained model with Davit-S as the backbone. The ratio calculates the percentage of efficiency metric compared to the fully fine-tuned baseline. Notations: 'Ro.' for Router, 'Ex.' for expert(s), $\theta$ is a threshold on the frequency used for an expert. We have two hybrid models: 1) 'Hybrid-A' directly combines 'Ro. w/ 1 Ex.', 'Prune 2/3 Ex.', and 'Top-K=2'. 2) 'Hybrid-B' combines 'Ro. w/ 2 Ex.', 'Prune 2/3 Ex.', and 'Top-K=3'.

| Method | Train. Par.(M) | Model Par.(M) | FLOPs (G) | Ratio | P365 top-1 | iNat18 top-1 | Pets top-1 | CUB top-1 | Cars top-1 | PASC. $mAP$ | City. $mIoU$ | NYU $mIoU$ | Mean |
|---|---|---|---|---|---|---|---|---|---|---|---|---|---|
| FT-Full | 88.9 | 88.9 | 9.2 | - | 59.0 | 72.9 | 94.0 | 88.2 | 95.0 | 78.6 | 81.4 | 77.4 | 79.9 |
| Adapter (Houlsby et al., 2019) | 14.8 | - | - | 16.6% | 50.7 | 62.4 | 81.1 | 75.8 | 80.8 | 67.7 | 69.9 | 66.8 | 68.7 |
| Ro. Only | **0.4** | - | - | 0.4% | 52.1 | 64.2 | 83.3 | 77.9 | 78.2 | 69.6 | 71.8 | 68.7 | 70.3 |
| Ro. w/ 1 Ex. | 5.4 | - | - | 6.1% | 57.4 | 70.7 | 91.3 | 85.8 | 94.7 | 76.5 | 78.8 | 75.2 | 77.8 |
| Ro. w/ 2 Ex. | 10.4 | - | - | 11.7% | 58.8 | 72.7 | 94.0 | 87.8 | 95.0 | 77.9 | 80.7 | 76.7 | **79.4** |
| Prune $\theta = 1\%$ | - | 60.2 | - | 67.7% | 58.9 | 72.8 | 93.9 | 88.1 | 95.0 | 78.6 | 81.4 | 77.3 | **79.9** |
| Prune $\theta = 5\%$ | - | 54.4 | - | 61.2% | 58.8 | 72.7 | 93.8 | 88.0 | 94. | 78.4 | 81.4 | 77.2 | 79.7 |
| Prune 1/2 Ex. | - | 59.9 | - | 67.3% | 58.8 | 72.8 | 93.9 | 88.0 | 93.9 | 78.6 | 81.4 | 77.3 | 79.8 |
| Prune 2/3 Ex. | - | **49.9** | - | 56.1% | 58.8 | 72.6 | 93.6 | 87.8 | 93.8 | 78.6 | 81.3 | 77.2 | 79.7 |
| Top-K=3 | - | - | 7.7 | 83.7% | 58.8 | 72.5 | 93.3 | 87.3 | 94.9 | 77.3 | 80.1 | 76.3 | **79.0** |
| Top-K=2 | - | - | 6.2 | 67.4% | 58.1 | 70.7 | 91.9 | 86.2 | 92.0 | 74.9 | 77.6 | 73.7 | 76.8 |
| Top-K=1 | - | - | **4.7** | 51.0% | 48.5 | 59.9 | 77.3 | 72.4 | 77.4 | 64.3 | 66.6 | 63.3 | 65.4 |
| Hybrid-A | 5.4 | 49.9 | 6.2 | - | 58.0 | 70.6 | 91.1 | 85.8 | 94.7 | 76.3 | 78.5 | 73.2 | 77.4 |
| Hybrid-B | 10.4 | 49.9 | 7.7 | - | 58.8 | 72.4 | 93.3 | 87.2 | 94.9 | 77.1 | 79.9 | 76.2 | **78.8** |

## 4.2 LegoNet is an Efficient Adapter

**Efficient in training parameters.** LegoNet can adapt quickly to a new task or dataset by tuning the router with a few optional experts and learning a new task head. During this process, all other parameters are frozen. The optional experts to be fine-tuned are randomly selected. Randomly selected experts perform similarly to selecting the expert with the highest or lowest use frequency on the downstream dataset (see Supp. Mat.).

In Tab. 4, our method is referred to as 'Ro. Only', 'Ro. w/ 1 Ex.', and 'Ro. w/ 2 Ex.', referring to tuning routers only, and routers with 1 or 2 experts per MoE module. We compare our efficiency in training parameters with the commonly used adapter (Houlsby et al., 2019), which adds an adapter module after each MoE MLP block. In contrast, we only need new lightweight routers (0.4M) and 1 or 2 additional experts per MoE module. Even updating only new routers outperforms the adapter baseline, and Ro. w/2 Ex. has performance close to the fully fine-tuned baseline. See Fig. 3.

**Dynamic scaling down.** Regarding model capacity, LegoNet can remove experts after learning a new router on a new task by removing least-used experts, followed by fine-tuning the entire model. We explore two methods of pruning: 1) Removing a few experts from each MoE layer. In Tab. 4, we attempt to remove 1/2 experts and 2/3 experts. 2) Removing all experts whose use frequency is lower than a threshold $\theta$ on the downstream dataset. This approach may result in a different number of experts in each MoE layer, but it has comparable efficiency to the first pruning method. See Tab. 4 and Fig. 3 for results and a comparison. This way of dynamic scaling model size can stabilize performance compared to the original model.

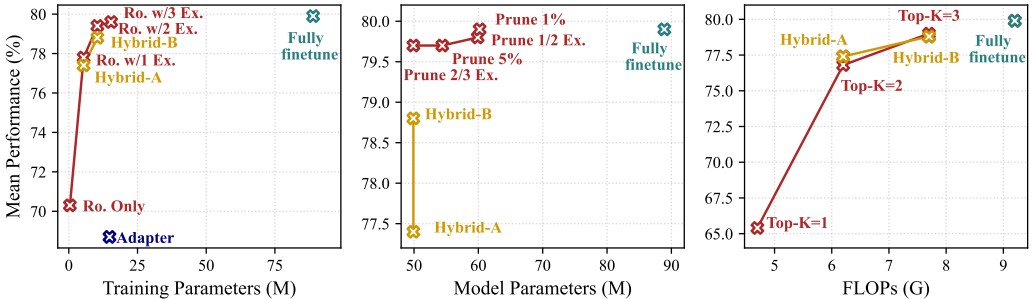

Figure 3: **Trade-off between efficiency and performance.** We visualize the trade-off between performance and training parameters, model parameters, and computation cost respectively.

Table 5: **Continual learning.** We conduct continual learning on these datasets one by one after pre-training and report final performance. All experiments are based on LegoNet with a DaviT-S backbone. The number of training and newly added parameters in the backbone per task are measured. 'Mean' is the mean performance on all datasets.

| Method | New params per task (M) | Train. params per task (M) | P365 top-1 | iNat18 top-1 | Pets top-1 | CUB top-1 | Cars top-1 | PASC. mAP | City. mIoU | NYU mIoU | Mean |
|---|---|---|---|---|---|---|---|---|---|---|---|
| LWF (Kirkpatrick et al., 2017) | 0 | 88.9 | 46.2 | 57.0 | 73.5 | 70.6 | 75.5 | 62.7 | 71.1 | 68.9 | 65.7 |
| Rou. only | 0.4 | **0.4** | 52.1 | 64.2 | 83.3 | 77.9 | 78.2 | 69.6 | 71.8 | 68.7 | 70.7 |
| Rou. w/ 1Ex. | 5.4 | 5.4 | 57.6 | 70.8 | 91.3 | 85.9 | 94.7 | 76.8 | 79.0 | 75.6 | 79.0 |
| Rou. w/ 2Ex. | 10.4 | 10.4 | 58.8 | 72.8 | 94.5 | 88.0 | 95.0 | 78.1 | 80.7 | 76.9 | **80.6** |
| FT-Full | – | – | 59.0 | 72.9 | 94.0 | 88.2 | 95.0 | 78.6 | 81.4 | 77.4 | 80.8 |

**Computational efficiency.** Much pre-training uses a relatively large backbone, but downstream tasks/datasets may not require such large model capacity. LegoNet can regulate the computation cost by learning new routers with a reduced Top-K. This gives a trade-off between performance and computation (see Fig. 3). For some datasets (*e.g.*, P365), it can achieve a relatively low computation cost (*e.g.*, 67.4%) while maintaining the same level of performance (*e.g.*, <1% drop).

**Combine all efficient adapting.** To further improve efficiency, the efficient adapting techniques can be combined. In Tab. 4, for Hybrid-B, we first learn a new router and remove 2/3 experts. Then, we fine-tune the router with Top-K as 3 along with two experts per module. This approach achieves a mean performance of 78.8, which is only 1 point lower than fine-tuning the entire model. Moreover, this method reduces training parameters, model parameters, and computation costs simultaneously.

### 4.3 CONTINUAL LEARNING.

Continual learning without any forgetting is achievable with LegoNet by learning new routers (0.4M) and a few optional experts on the new dataset. We compared it with the common regularization-based continual learning baseline LWF(Kirkpatrick et al., 2017). As demonstrated in Tab. 5, LegoNet has three significant advantages: 1) No forgetting on the learned datasets. 2) Only a smart part of the model needs to be trained on new datasets, requiring only 10.4M training parameters, while LWF needs to tune the whole model (88.9M). 3) Comparable performance to fully fine-tuning the whole model on every dataset. These results further prove the effectiveness of LegoNet as a general MTL framework to handle an ever-increasing number of tasks.

## 5 CONCLUSION

Our study focused on a scalable multi-task model that can piece together sub-networks from single-task models and its ability to adapt to downstream datasets. Experiments are conducted to demonstrate its effectiveness, dynamic scaling property, and adaptability. The broader impact of our work could be significant in terms of advancing scalable MTL and effective adaptation of large-scale models. One limitation of LegoNet is it may be biased toward certain datasets and require more training iterations for convergence compared to single-task models.

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

# A APPENDIX

Table 6: We compare three ways of selecting a subset of experts to fine-tune, while freezing the remaining experts. We first learn new routers on the new downstream to determine each expert's frequency of being chosen. Random represents randomly choosing experts. Best represents choosing the experts with the highest frequency. Worse represents choosing the experts with the lowest frequency. We report mean top-1 accuracy on CUB, Cars, and Pets. Other settings are the same as in Table.3 in the paper.

|  | Random | Best | Worse |
|---|---|---|---|
| Ro. w/1 Ex. | 90.6 | 90.5 | 90.6 |
| Ro. w/2 Ex. | 92.3 | 92.3 | 92.2 |

## A.1 DIFFERENT WAYS TO SELECT EXPERTS TO BE FINE-TUNED.

Tab. 6 compares various methods of selecting experts to fine-tune while freezing the rest. We compare random selecting experts and selecting experts that are more or less likely to be chosen by routers. We find out that the selection method does not significantly affect the fine-tuning performance. Therefore, we use random selection for simplicity.

## A.2 ABLATION ON TOP-$K$.

As shown in Tab. 7, we explore the effect on Top-$K$ in MoE module. The experiment setting is the same as in Tab.1 in the paper with 12 experts per MoE module. We report the mean performance on pre-train and downstream datasets of our MHTL with Davit-T as the backbone. To control the FLOPs to be the same for different Top-$K$, the hidden dimension of MLP experts is divided by $K$. All experiments have the same parameter size and the same FLOPs. We find that Top-$K = 4$ has the best performance.

## A.3 ABLATION ON THE NUMBER OF EXPERTS.

As shown in Tab. 8, we explore the effect on number of experts $E$ for MoE MLP layer. The settings are the same as in A.2 with a Top-$K$ as 4.

Table 7: **Ablation study of Top-$K$ on MoE MLP layer.**

|  | FLOPs(G) | Params(M) | Hidden Dim | Pre-train mean | Downstream mean |
|---|---|---|---|---|---|
| K=2 | 5.1 | 51.2 | 768 | 58.1 | 80.3 |
| K=4 | 5.1 | 51.2 | 384 | 58.2 | 80.4 |
| K=6 | 5.1 | 51.2 | 256 | 57.9 | 80.0 |

Table 8: **Ablation study of expert number $E$ on MoE MLP layer.**

|  | FLOPs(G) | Params(M) | Pre-train mean | Downstream mean |
|---|---|---|---|---|
| E=6 | 5.1 | 33.4 | 57.2 | 78.5 |
| E=9 | 5.1 | 42.3 | 57.9 | 80.0 |
| E=12 | 5.1 | 51.2 | 58.2 | 80.4 |
| E=15 | 5.1 | 60.1 | 58.2 | 80.5 |

## A.4 TRAINING DETAILS

**Optimization and convergence.** Each task in our framework has a dedicated module and its own loss. The losses on datasets $D_i$ are weighted and alternately optimized with predetermined weights $w_{l_i}$. Gradient conflicts between tasks pose a challenge, slowing convergence. Well-defined loss and sampling weights contribute to training stability, and the large batch optimizer Lamb (You et al., 2019) is effective in heterogeneous training. Convergence in this setting typically requires approximately 50% more iterations than single-task training due to the complexity of joint optimization. Loading pre-trained single-task models can significantly accelerate training, as discussed in the next section.

**Training details.** During pre-training, data sampling weight is set to {3, 2, 1}, loss weight is set to {1.0, 0.6, 0.2}, and batch size is set to {64, 2, 2} for classification, detection, and segmentation, respectively. Weight decay is set to 0.05 and the maximal gradient norm is clipped to 0.1. We use a simple triangular learning rate schedule with a maximum learning rate of 0.004, as in DaviT.

