# OpenReview forum: "LegoNet: Piecing Together and Breaking Apart Sub-Networks for Scalable Multi-task Learning"
_ICLR.cc/2024/Conference — ICLR 2024 Conference Withdrawn Submission_

### Official Review · Reviewer_MPaF · 2023-10-28

**Soundness:** 2 fair
**Presentation:** 3 good
**Contribution:** 2 fair
**Rating:** 5
**Confidence:** 5

**Summary:**

The paper introduces an innovative approach to multi-task learning in computer vision. It advocates for task-specific modules rather than a one-size-fits-all approach, and presents LegoNet, a framework that assembles smaller sub-networks for different vision tasks, much like assembling Lego pieces. LegoNet demonstrates efficiency and comparable performance to single-task models on established datasets. It also showcases adaptability, allowing dynamic sub-network assembly and disassembly, fine-tuning with fewer parameters, and reduced computational demands. This paper offers a valuable contribution to the field by bridging single-task and multi-task learning in a flexible and adaptable manner.

**Strengths:**

1. **Innovative Approach**: The paper presents an innovative approach of initializing the Mixture-of-Experts (MOE) network with pretrained weights from single-task models. This concept, while not entirely novel, offers practical and potentially effective solutions for multi-task learning.

2. **Performance Enhancement**: The paper's approach demonstrates improved performance, a critical metric in practical machine learning applications. Enhanced model efficiency and effectiveness contribute significantly to the paper's strength.

3. **Applicability**: Leveraging prior knowledge through pretrained weights holds practical relevance. This approach can enhance multi-task learning models' real-world applicability and streamline their development.

**Weaknesses:**

1. **Lack of Technical Contribution**: The core novelty of the paper is seen as its utilization of pretrained weights from single-task models to initialize the Mixture-of-Experts (MOE) network. However, this idea is considered trivial because the MOE system itself does not introduce new technical contributions. Loading better pretrained weights may not be a strong novelty, as it doesn't significantly advance the field.

2. **Lack of Originality**: The concept of building modular neural networks and reassembling them, akin to Lego blocks, is not entirely novel. Those ideas are introduced in back is already in the literatures [A,B,C], but are not properly reference and discussed in this paper.

3. **No throughout Analysis**: While the idea of composing specific expert is valued, the paper falls short in providing a clear understanding of the relationship between pretrained single-task networks and the final model. It remains unspecified which twelve single-task models have been incorporated (a detail omitted throughout the paper). Additionally, there is a lack of clarity regarding the selection or pruning of submodels for specific downstream tasks.

[A] Ye, Hanrong, and Dan Xu. "TaskExpert: Dynamically Assembling Multi-Task Representations with Memorial Mixture-of-Experts." Proceedings of the IEEE/CVF International Conference on Computer Vision. 2023.
[B] Yang, Xingyi, et al. "Deep model reassembly." Advances in neural information processing systems 35 (2022): 25739-25753.
[C] Yang, Xingyi, Jingwen Ye, and Xinchao Wang. "Factorizing knowledge in neural networks." European Conference on Computer Vision. Cham: Springer Nature Switzerland, 2022.

**Questions:**

While the obtained performance is promising, I am particularly interested in understanding the relationship between the initially loaded single-task experts and the final required expert. This curiosity is heightened by the author's suggestion of pruning unused experts. How does this relationship manifest in practice? Is there an intuitive connection, such as using a segmentation model for a segmentation task?

---

### Official Review · Reviewer_pEdg · 2023-10-30

**Soundness:** 3 good
**Presentation:** 3 good
**Contribution:** 2 fair
**Rating:** 5
**Confidence:** 4

**Summary:**

This paper proposes multi-task mixture-of-expert architecture LegoNet for computer vision tasks with multiple datasets: classification, semantic segmentation and detection. The architecture can be built on top of a backbone architecture, e.g. SwinTransformer, and then any single task learning architecture built with the same backbone can be merged into LegoNet by expanding with additional experts. Since the network encourages multiple datasets / tasks to share the similar features to obtain a sparsely activated network, it naturally inherits some interesting properties: e.g., efficient adaption to new tasks by just learning a new router / gating network, direct support for continual learning since the network can easily store the gating weights for each task. As a result, LegoNet achieves competitive results on the same model architecture but without pre-training.

**Strengths:**

1.	The paper is well-written, and all the learning components are well-explained and supported by a sufficient number of experiments.
2.	The most interesting part of this paper is the discussions of different use cases such as for parameter-efficient transfer learning and continual learning, which I think could be expanded a bit more. See weaknesses.

**Weaknesses:**

In general, the paper itself is quite self-sufficient. But I hope the authors could answer and consider these following comments hopefully to further improve the paper.

1.	The novelty. The concept of applying MoEs to multi-task learning is not new. I am still a bit confused about how the proposed method is different from other multi-task MoE architecture, including the heavily compared Mod-Squad architecture. Also, I would love to hear how this paper differentiates itself from other recently proposed multi-task MoE architectures: AdaMV-MoE (Chen et al), TaskExpert (Ye et al) and DSelect-k (Hazimeh et al). I think a small section or a paragraph highlighting the differences and new discoveries are important, e.g. this paper focuses more on architecture modification in transfer learning and continual learning settings (as said in the related work section). But do these baselines not have this property? How about comparing other parameter-efficient baselines like LoRA? So in general I would suggest the paper better position itself with a clear objective: 1. If it’s to propose a new architecture, then the experiment should focus on comparing with sota single and multi-task architectures; 2. If it’s to find good properties in MoE architecture in multi-task learning, then most experiments should be centred around transfer learning and continual learning, and argue why the proposed strategy is better and more efficient.

2.	Weight inheriting from single-task networks. In Table 2, the paper shows inheriting weights from a single-task model can improve training efficiency. Are these network weights from pre-training with ImageNet21k or task-specific fine-tuning?  If it’s true pre-training weights, is it more efficient than standard task-specific fine-tuning in a single task network or standard parameter efficient fine-tuning with linear probe or LoRa / Adaptors? Table 4 explores this setting, but are all methods applied on LegoNet instead of the single-task network? If a single-task pre-trained network can also efficiently and effectively adapt to other downstream tasks (as commonly evaluated in the self-supervised learning literature), the value of this paper needs to be rethought.

I would consider raising my rating if the author provides an adequate rebuttal.

**Questions:**

See weaknesses.

---

### Official Review · Reviewer_thbj · 2023-10-30

**Soundness:** 3 good
**Presentation:** 3 good
**Contribution:** 3 good
**Rating:** 8
**Confidence:** 4

**Summary:**

The paper aims at solving the multi-task learning problems by designing a flexible and extendable pipeline. More specifically, the authors hypothesize that each vision task requires its specific designed module to use different forms of perception and they present the LegoNet. The LegoNet built on all pre-trained single-task learning networks by learn the Mixture-of-Expert layers to fuse the features from multiple experts loaded from single-task learning networks. The model can be extended to new tasks by learning new expert selecting routers or add/delete experts. Extensive experiments are performed to verify the effectiveness on multi-task learning, transferring to downstream problems and continual learning.

**Strengths:**

* The paper focuses on a practical problem.

* The proposed pipeline is flexible and extendable to downstream tasks and continual learning setting.

* The proposed method is extensively verified on different settings.

* In general, the paper is well-written and easy to follow.

**Weaknesses:**

* One of the limitation of the pipeline is that it can be very costly when there are myriad tasks to perform in pretrained stage. In this case, it maybe impossible to load all tasks weights into the LegoNet even using part loading.

* Another limitation of the pipeline is that it may not fully leverage cross-task relations for learning the models compared with learn one network from multiple tasks as learning one network from multiple tasks would enable the knowledge to transfer across tasks.

* Some related literature is missing. [A][B][C] and it seems that the results of the hard-sharing are missing in Table. 1 as well.

* Only one PEFT method (Adapter) is compared and the comparisons of more methods (such as LoRA) are suggested.

* The method requires 96 Tesla V100 GPUs for training which is costly.

**Questions:**

* There are two related papers that leverage the single-task learning network to build the multi-task learning network and transfer/adapt to downstream tasks, which are missing in this work. For example, [A] proposes learning the hard sharing network by distilling the knowledge from multiple single-task learning networks into it. Another method [B] is also very related to this paper. [B] learns to combine features from different single-task learning networks for the downsteam tasks.

* The method is shown that it can be adapted to downstream problems by fine-tuning the experts selecting routers or adding new experts. And usually when a task that is very different from the source task and is sampled from a very different domain (dataset), it usually requires a more significant adaptation, e.g. adding new experts while a task that is very similar to the pre-trained ones may only need fine-tuning the experts selecting router. Then my question is how to determine whether adding experts or not and how many experts should be added.

* The loading strategies assume that the number of experts in the multi-task network is the same as the total number of experts in single-task learning networks. How about using less number of experts when using partial loading. For example, if we randomly select 3 experts from each single-task network and there are 3 tasks, how about the results of using 9 experts in multi-task learning network?

* Why does LegoNet which loads all single-task learning models which contains all single-task learning networks' parameters, have less parameters than the ones of all single-task learning network?

* Are the experts selected by filtering the outputs from the unselected experts? If so, then the computational cost increase when the total number of experts increases as we will need forward the samples to all experts before applying the gating function to select the outputs.

* The method is quite related to (He et al., 2022; Ghiasi et al., 2021), how well is the proposed method compared to them, especially (He et al., 2022)?

[A] Li et al., Universal Representations: A Unified Look at Multiple Task and Domain Learning, IJCV 2023.

[B] Dvornik et al., Selecting Relevant Features from a Multi-domain Representation for Few-shot Classification, ECCV 2020.

[C] Rebuffi et al., Learning multiple visual domains with residual adapters, Neurips 2017